# A Case Study of Active Aging through Lifelong Learning: Psychosocial Interpretation of Older Adult Participation in Evening Schools in Korea

**DOI:** 10.3390/ijerph18179232

**Published:** 2021-09-01

**Authors:** Ilseon Choi, Sung Ran Cho

**Affiliations:** 1Graduate School of Education, Global Campus, Kyung Hee University, 1732 Deokyoungdaero, Giheung-gu, Yongin-si 17104, Korea; education@khu.ac.kr; 2Graduate School of Education, Hongik University, 94 Wausan-ro, Mapo-gu, Seoul 04066, Korea

**Keywords:** active aging, lifelong learning, evening school, equivalency examination, motivation, goal

## Abstract

Lifelong learning is a key element of the conceptual framework of active aging. To understand how older adults experience active aging through participation in lifelong learning, the authors conducted a qualitative case study. The research participants were older adult learners attending evening schools aiming to pass the equivalency examination. Data were collected primarily using semi-structured interviews with five older adult learners, and additional data were collected from relevant documents. Data analysis and thematic discussion provided insights into how older adults experience active aging by participating in lifelong learning. Data analysis identified themes of overcoming limited education, taking the equivalency examination, and evolving goals. Thematic discussion revealed that older adults began learning to meet deficiency needs; however, they developed their goals after attending evening schools and passing the equivalency examination. In addition, lifelong learning is an indispensable element of active aging not only because learning is good for older adults’ wellbeing, as reported in the literature, but also because older adults become more active in the systemic change of their environment and in the setting goals for their lives.

## 1. Introduction

The conceptual framework of active aging has been a 21st century global approach to older adult wellbeing. According to the World Health Organization (WHO), active aging refers to “the process of optimizing opportunities for health, participation, and security in order to enhance quality of life as people age” [1]. Lifelong learning in old age deserves attention in the active aging framework because “participation” refers to social, cultural, and civic activities, and lifelong learning is essential to facilitate such activities. Recently, the WHO added lifelong learning as the fourth element of the active aging framework and refined the notion of active aging as the “process of optimizing opportunities for health, lifelong learning, participation, and security in order to enhance the quality of life as people age” [2]. In this context, active aging that includes lifelong learning as one of its constituents has been adopted as a key aspect of aging policy in many countries, including Korea [3,4]. 

Lifelong learning and its role as a catalyst for a healthier and more socially engaged life in old age has been documented in many research articles [5]. In brief, older adult learning was found to play a key role in health and psychological wellbeing [4,6]; gaining new knowledge and skills for adjustment to old age and the rapidly changing world [7,8]; helping older adults to acquire social and economic resources [9,10,11]; and improving social relations with same-aged or older/younger peers [12,13].

In the study of older adult participation in learning activities, motivation has been a major issue. This is because motivation and participation are closely related, and participation is assumed to be based on motivation [14]. Since McClusky’s [15] pioneering study, there have been empirical studies examining older adults’ motivations for participating in learning, including those of Schaie and Willis [16]; Kim and Merriam [17]; Boulton-Lewis, Buys, and Lovie-Kitchin [18]; Findsen and McCullough [19]; Chen and Wang [20]; and Xiong and Zuo [21]. However, motivation has been seen as a psychological construct to explain human activity through internal processes of thinking, reasoning, and feeling, etc. [14,22]. Inherent in this construct is the notion that an internal drive mechanism triggers participation in activities, and people meet their needs by participating in social activities. Sociologists, on the other hand, have explained motivation in relation to social factors. Woldkowski [23], under a sociological perspective, examined adults’ motivation to learn and argued that social factors such as unemployment and schooling “powerfully affect the consideration of formal learning for adults”.

The sociological perspective has been helpful in compensating for the shortcomings of understanding older adults’ participation through psychological constructs. Thus, some suggested a combined approach to human behavior incorporating both the psychological and sociological perspectives. While discussing the barrier to participation in adult learning, Rubenson points out that we need to include structural factors and analyze the interaction between them to allow interpretation of the individual conceptual apparatus [24]. In his study of older adult education, Londoner provided a model of “psychosocial interpretation of adult participation” [22]. As the title denotes, he assumes that older adult learners are motivated toward specific goals based on social settings and psychological needs. 

The aim of this study was to gain in-depth understanding of Korean older adult learners participating in evening school with the goal of passing equivalency examinations. We used Londoner’s psychosocial model to analyze the social setting of evening school and the needs of learners [22]. Based on the findings and discussion, we provide insights into how older adults experience active aging through participating in lifelong learning.

## 2. Methods

### 2.1. Study Design

This research used a qualitative case study design to examine older adults’ participation in evening school. This type of study originated from field research methods of cultural anthropology in the beginning of the 20th century and developed its procedures for data collection and analysis under the influence of sociologists at the University of Chicago in the 1920s through the 1950s [25]. From Stake’s [26] methodological typology, which considers the instrumental case study and the intrinsic case study, we adopted the former, which seeks to obtain general understanding of a case rather than satisfying just intrinsic interest. Furthermore, we adopted the “single-case design” [27], which refers to investigation of a single phenomenon (i.e., participation in an equivalency examination preparatory program at an evening school) as experienced by multiple participants. We selected five research participants from four evening schools and collected data for two months at the end of 2020. To understand the phenomenon of concern, we used three temporal categories of investigation of past, present, and future experiences of participants.

### 2.2. Context

Evening schools in Korea are non-formal schools outside the public education system. Thus, students of these schools who want formal recognition of their education need to pass an equivalency examination. Since the amendment of the Lifelong Education Act in 2008, evening schools have been categorized legally as literacy education institutions and are funded by local government subsidy. Most of the current evening schools were found in the 1970s, and the students at the time were mainly poorly educated working youths. Since then, as the provision and enrollment of public schools have increased (for example, 73.3% middle school enrollment in 1980), the majority of evening school attendees has shifted from teenagers to middle-aged women. Since the 2000s, the number of older adult learners has increased, with adults in their 50s and older accounting for the majority (71.7%) of evening school students [28]. As of 2020, there were 45 evening schools in Korea, and the average number of students per institution was 56. Among them, roughly 20%, 40%, and 40% of the students participate in preparatory courses for the Elementary School Equivalency Examination (ESEE), Middle School Equivalency Examination (MSEE), and High School Equivalency Examination (HSEE), respectively. The four involved evening schools are large and well known in Korea. Three of them are located in metropolitan areas, and the other one in the central region of Korea. The schools were established in 1982, 1963, 1987, and 2005, respectively, and the number of registered students is 190, 75, 137, and 120, respectively. All of the schools are private institutions managed by volunteer teachers and are financially supported by individual donation and educational subsidy.

The equivalency examination, which is the main goal for evening school learners, is an academic recognition system that provides an elementary, middle, and/or high school diploma through examination. In the incipient period in 1950’s, the examination was performed mostly by teenagers who had not attended formal school. Since 1980, the participants have diversified to include adults in addition to school dropouts. In 2019, 4,167 applicants took the ESEE, 11,270 took the MSEE, and 43,816 took the HSEE. For the HSEE, adults aged 50 or older accounted for 12.6% of all examinees [29]. Internationally, the MSEE and the HSEE in Japan are similar systems to those in Korea, with the exception that Japan’s examinations grant higher school entrance qualifications but do not grant diplomas, as is the case in Korea. The General Educational Development (GED), implemented in the United States and Canada function similarly to the Korean HSEE, but the GED has the distinct characteristic of being an adult-only academic recognition system, given that it provides college entrance qualification to people aged 19 and older, although that varies slightly from state to state [30].

### 2.3. Study Participants

To best understand the phenomenon of interest, we purposefully selected five research participants who are “information rich” [31], according to the recommendation of the principals of four representative evening schools. The participants were aged between 59 and 75 years, with four being female and one being male. Pseudonyms were used to identify participants: Shin, Ahn1, Ahn2, Chung, and Soo. As an elementary school graduate, Shin passed the MSEE and is currently preparing for the HSEE. As an elementary school dropout, Ahn1 passed the ESEE and is currently preparing for the MSEE. As a middle school dropout, Ahh2 passed the MSEE and HSEE and is currently attending an online college. As an elementary school dropout, Chung passed the three equivalency examinations successively and is currently attending a university. Soo had no education before she attended the evening school, but she passed the three examinations, entered a college, and has since graduated.

### 2.4. Data Collection

We carried out semi-structured interviews with the learners in October and November 2020. All interviews were conducted individually for two to three hours and were audio recorded and transcribed in their entirety. The interviews, consisting of open-ended questions, were designed to prompt the interviewees to think about their experiences with evening school programs and the equivalency examinations. The following are examples of questions asked/statements to each of the participants: What did drive you to attend evening school; Please tell us about your learning experiences in evening school; Please tell us how knew about the equivalency examination and what motivated you to take it. For the questionnaire, author A designed the draft, author B developed it to the final version, and both participated in the interview. Documents were collected before and after the interviews. They included information from the evening schools’ websites, program orientation books, brochures and handouts about the equivalency examinations, program curriculums, and newsletters from the schools.

### 2.5. Data Analysis

We started data analysis simultaneously to data collection. We recorded analytic notes during the interviews and documentation review. After data collection, we conducted thematic analysis to generate codes using “theory-driven code development” as suggested by Boyatzis [32]. Both authors participated in the analysis. For internal consistency of researchers, we conducted thematic analysis individually based on Boyatzis’ method, compared the themes, and searched for similar codes. Author A developed draft thematic codes, which were reviewed by author B and adjusted to the final thematic codes. The thematic codes were built on Londoner’s [22] temporal lifeline model of needs → social system → goal gratification, a model of older adults’ educational participation depicted on Figure 1. According to Londoner’s model, present human behaviors are exhibited within social systems, past needs trigger the present behavior, and future goals are achieved by participating and interacting with others in a social system. In the model, needs and goals are categorized as instrumentally or expressively oriented, according to whether their gratifications are delayed or immediate. In addition, needs and goals are related logically to participatory behavior of a person; for example, instrumentally oriented needs are meaningful only when they are linked with the person’s instrumentally oriented goals. The model provided insights to analyze the data focusing on instrumentally oriented needs and goals. Throughout the analysis, the following thematic categories were identified: overcoming limited education, taking the equivalency examination, and evolving goals. Themes were induced mostly from the analysis of interview data. On the other hand, documents were used to guide the interviews and to describe the context of the study regarding evening schools and equivalency examination.

## 3. Results

### 3.1. Overcoming Limited Education: Inferiority Complex

There are various circumstances under which the research participants were poorly educated. Soo was unable to enter elementary school due to her father’s early death and poverty at home, was entrusted to another family at the age of nine, where she lived a “subhuman life,” and then worked in a factory as a teenager. After she got married, she was busy with her work and house chores, so she could not attempt studying. Ahn1 dropped out of elementary school because she was unable to adapt to changing schools due to her father’s frequent transfers for work, Chung dropped out of elementary school to help with housework, and Shin got a job right after she graduated elementary school due to family poverty. Like Soo, these individuals were unable to resume their studies even when they became adults due to work and family circumstances. Ahn2, the only male among the research participants, was not interested in studying and dropped out of middle school; he got a job in a factory and did not have a chance to further his education.

All of these participants suffered from an inferiority complex due to their limited education, and some attempted to overcome it. Shin felt deep sorrow about not going to middle school and said that she wished to view the contents of related textbooks before she died. When she was working as part of the culinary staff at a restaurant, she was ashamed of not being able to inspect food material written in English. At that time, she committed herself to learning English someday. Soo, who had no education, had been kicked out of school several times when she followed her village friends to elementary school. Ashamed of her lack of education, she hid it from her husband and children.


*My kids didn’t know that I was uneducated. I had learned how to read at church, so I could answer most of the questions my children asked when they went to elementary school. That’s probably why they didn’t guess that I was uneducated. My husband also didn’t know that I hadn’t even entered elementary school. I’ve never told him that.*
*(Soo)*

Ahn2 also hid his educational background from his daughters. After marriage, Ahn1 and Chung were both ashamed of themselves, being less educated than the wives of their husband’s colleagues. Thus, when they went to couples’ association meetings, they were intimidated due to an inferiority complex. To overcome their limited education, Ahn1 tried to take the equivalency examination by herself in her early 30 s, as did Ahn2 in his mid-20 s; however, both failed. Chung learned computer skills, Chinese characters, and accounting to run her flower shop, and Soo learned to read at church.

### 3.2. Taking the Equivalency Examination: Trials and Achievements

The research participants obtained information about evening school through various channels. Shin saw the sign of an evening school on her way to work but hesitated for several years before finally entering the school to study English. She enrolled in the equivalency examination preparatory program after learning about it during the English program at the evening school. She passed the MSEE on the first try but has failed the HSEE several times and is preparing for the next HSEE. Chung enrolled in evening school after hearing information about it from a friend. Soo learned about the evening school after being recommended by the director of her senior welfare center. She visited the school and joined the equivalency examination preparatory program the following day. Chung and Soo describe the day when they entered evening school:


*One of my friends stopped by the flower shop I used to run and said she was studying at evening school. I asked her what she was studying, and she replied that there is an evening school titled Open School where you can study elementary, middle, and high school programs, and she said it’s so fun to study. I thought that was the place I was looking for, so I searched for information about the school on the internet. I called and told them that I wanted admission counseling and asked for the location of the school. I visited and registered at the school and started studying the next day.*
*(Chung)*


*I visited the evening school at the recommendation of a welfare center director. I started interviewing the principal, and he asked me (to test my writing) to write my home address and name. During the meeting, I said that I hadn’t even entered elementary school. The principal recommended that I attempt the equivalency examination. I asked how to take the examination because I had never heard of it. The term was unfamiliar to me when I first heard it. The principal explained that it was a test to get a certificate for elementary school graduation. “Really? Then I’ll get started.” I began studying without knowing exactly what the equivalency examination was. I was nervous and worried about how to study in the beginning. That’s how I started studying, but I passed the ESEE in two months (after entering the evening school).*
*(Soo)*

Ahn1 happened to see an advertisement about the evening school’s equivalency examination preparatory program on a bus. On that same day, she visited the evening school, enrolled, and began participating in the program. She passed the ESEE and is currently preparing for MSEE. Ahn2 was introduced to the evening school by an evangelist at his church, passed the MSEE and HSEE, and went to an online college. Below is a recollection of the day when Ahn2 took the HSEE.


*You can get a rough score if you pre-score it. Based on the pre-scored results, I was sure that I had exceeded the passing score. I just cried for joy while driving home. I cried a lot. I called my brother and mother and told them that I thought I had passed the examination. My family knows that I have a deep sorrow regarding my limited education. I was satisfied and grateful to God that that sorrow was resolved. If I had ignored the recommendation to attend the evening school, I would have regretted it until now. I was thankful that I studied at evening school and passed the HSEE. In addition, I was really grateful that I achieved my goal (of getting a high school diploma) in such a short period of time.*
*(Ahn2)*

All of the research participants passed the equivalency examinations more easily than they expected when they began studying. One reason for this was that the exam-inations have continuously changed in favor of older learners. The changes in evening school and in the equivalency examinations in Korea can be cited as the reason why the research participants were able to easily adapt to the school and pass the examina-tions in such a short period of two or three years.

### 3.3. Evolving Goals: Addiction to Learning

After the research participants entered evening school and passed the equivalency examinations, they gradually began to set higher goals. Ahn2, who dropped out of middle school, entered evening school with the goal of obtaining a middle school diploma, but has since passed the HSEE and went on to college.


*At first, I didn’t have a strong expectation of passing the equivalency examination. But while studying here, I became greedy. At first, I didn’t think about going to college because it could be hard to study at night after work, but I changed my mind after passing the HSEE. I decided to go to college to gain more knowledge and be more sophisticated, and I looked for an online college so I can study while continuing to work. I looked for a major related to Christianity and found it at SS online college. So, I entered the college.*
*(Ahn2)*

Other research participants also set higher goals after they studied at evening school. Soo, who went to evening school with the goal of obtaining an elementary school diploma, has passed the MSEE and graduated from a junior college. Chung started studying at evening school with the goal of obtaining a high school diploma and is currently attending a university. After passing the high school equivalency examination, she dreamed of college to study Chinese literature further while studying Chinese characters at a lifelong learning center. She aimed for a junior college that was easier to enter but changed her goal to a four-year national university and was accepted to the university at the age of 74. Since she was the oldest freshman at the university, she became the talk of the town and was interviewed on a local television station, where she said her next goal was entering a graduate school. Ahn1, who is preparing for the MSEE, hopes to go to college if her health allows.


*I think learning is a bit addictive. At first, I aimed for an elementary school diploma; I thought I would be satisfied with it. However, once I passed the ESEE, I changed my mind to take the MSEE and HSEE and go to college.*
*(Ahn1)*

If research participants pass the equivalency examination, they will be able to resolve their inferiority complex and have a more positive attitude toward life. In the early days of evening school, Chung did not inform people around her that she attended an evening school, but after passing the MSEE and enrolling in the HSEE preparatory program, she proudly informed others that she attended the school. She also gained confidence after she enrolled at a university.


*I couldn’t tell others that I was going to evening school because it’s like disclosing the fact that I’m uneducated. Once I passed the MSEE and got promoted to the HSEE preparatory course, I changed my mind. From then on, I started to recommend my friends to come to the evening school and study together. I’ve gained confidence. Passing the equivalency examinations reminded me of the past that caused me to feel inferior, but I’m not ashamed of those things anymore. I’m not intimidated by others. You know, I have a higher education than them now. I go to a four-year university, not a two-year college, and furthermore it’s a national university. Now I have a sense of pride.*
*(Chung)*

Soo overcame the depression that had plagued her after she passed the HSEE and went to college at the age of 71. She is proud of herself for graduating from college and for her position as a reading instructor and “shows off” her achievement to her late mother in the form of a soliloquy. She wants to do well in her role as a reading instructor and perform activities that contribute to society. Ahn1, who is preparing for the MSEE, also said that, after finishing her studies, “[she wants] to do volunteer work that helps someone”. 

## 4. Discussion

### 4.1. Needs

Understanding adults’ educational needs has been a major concern of adult educators because the success of a program depends largely on effective needs assessment [20]. Londoner’s model intended to serve as a theoretical rationale for the needs assessment of older adults’ educational participation. He approached the needs considering “the social setting and the older person’s needs jointly” [22] and portrayed the model as a temporal line stretching from the past throughout the life span. According to the model, the learning needs of older adults emerge from a “causal condition,” an educational deficiency based on their life history. This demonstrates what Maslow [33] called deficiency of needs. The research participants’ inferiority complex, caused by limited education from their childhood, had a negative impact on their adult lives, and their educational background was a disgrace to be hidden from others, including their children and spouses. To resolve the complex, Ahn1 and Ahn2 attempted to take the equivalency examination by themselves but failed. They participated in the equivalency examination preparatory programs offered by evening schools as soon as they visited the schools and learned about the programs.

Many researchers, including McClusky [15], Houle [34], Eisen [35], and Londoner [22] have attempted to categorize older adults’ learning needs. According to McClusky’s five categories, needs of the research participants would fall under the coping needs category, which emphasizes remedial education to overcome any deficiency in formal childhood education. The needs are similar to those of Houle’s goal-oriented learners who use education as a means of achieving other goals. Their needs are Londoner’s instrumentally-oriented needs in that their gratification is delayed for other goals. The goals of the learners in this study are similar to those of learners in the credentialing program of Eisen’s four-part typology. Credentialing programs usually are delivered in a conventional classroom setting, with set meeting times, to groups of older adult learners who are interested in earning a credential, such as a general education diploma. Similar to credentialing programs, the research participants studied at non-formal evening schools with a goal of earning the formal accomplishment of a diploma.

### 4.2. System

According to Londoner’s model [22], the social system is located between past needs and future goals. The present social system consists of the social situations within which people’s patterns of actions and interactions are analyzed. Evening school and the equivalency examinations were the social systems examined in this research. Both are involved in the non-formal learning context and the formal recognition of its results. All learning takes place in a social context and often is divided into formal, non-formal, and informal settings [36,37]. Formal learning here refers to learning activities that take place in educational systems and often lead to degrees or credits. On the other hand, non-formal learning refers to activities outside educational systems, while informal learning refers to everyday life experiences from which we learn. Among the three types of learning, recognition of non-formal and informal learning outcomes has been the focus of lifelong learning, and many countries have developed recognition systems based on their historical, political, and sociocultural background [38]. Due to the characteristics of being government-led and educational credentialism-oriented, Korea’s recognition systems are operated around academic recognition organized by the Ministry of Education. This differs from the American and Canadian models, where universities conduct the recognition system autonomously, or the Australian model, which centers on labor market utilization of competencies [39]. Korea’s system of equivalency examinations recognizes the results of learning in the non-formal setting of evening schools as the academic background.

Equivalency examinations in Korea have continued to change since their inception. Favorable changes that have a significant impact on the increase in older adults’ participation in equivalency examinations are a decrease in the difficulty of the examination and abolition of the minimum passing score for each subject. Due to the changes, the pass rate of the HSEE, for example, has increased from 33.8% in the 1990s when Ahn1 and Ahn2 tried the exam to 71.22% in 2019, and the minimum limit was abolished in 2004 [40,41]. This change is the result of the government’s acceptance of continued requests from evening school learners, teachers, principals, and literacy education activists to change the equivalency examination to be adult-friendly. The change of system/structure and the role of actors/agency in the change are continuing issues of social theory. For example, Giddens’ [42] “structuration” theory is based on the idea that structure is both input to and output of human actions. In the field of adult education, Darkenwald and Merriam [43] argued that barriers to participation can be conceptualized as a continuum on the structure/agency scale, and Jamieson [44] found that individuals become less influenced by structural factors and more agentic with age. In Korea, changes in the equivalency examination system continue at the request of those involved in the examination. For example, the argument for separate equivalency examinations for teenagers and adults is ongoing [28], and it is expected that the examination will become friendlier to older adult learners.

### 4.3. Goals

According to Londoner [22], people are not only pushed by some psychological needs, but are also pulled toward some specific goals. Among motivation theories, goal theories explain how people’s behavior depends on the goals they set for themselves [45]. Londoner divides goals into expressively and instrumentally oriented ones: the former seeks immediate gratification from the behavior itself, and the latter delays immediate gratification in the interest of larger gains derived from achieving goals. The goals the research participants seek are instrumentally oriented in that they discipline themselves to renounce immediate gratifications in the interest of gains from passing the equivalency examination. For example, Shin says, “Refusing friends’ temptation to go hiking or play, I am concentrating on studying for the examination.” According to Londoner, this temptation causes a dilemma to face choices in a variety of options. The choice is between immediate gratification and delayed gratification. In this dilemma, Shin chose the latter. This study showed that older adult learners do not simply delay gratifications, they develop and pursue goals. While attending evening school, Ahn2, Soo, and Chung set a goal of going to college, and after passing the equivalency examination, they went to college; Soo and Ahn1 developed a goal of volunteering to help society after passing the examination.

To describe the development of goals of the research participants, we used the title “evolving goals,” which the authors borrowed from NG [46]. He used the term to explain “the developmental nature of motivation” of older adult learners and added that the notion of social embeddedness is important to understand motivation. The social embeddedness refers to “collective expectation” [22], such as diplomatism and the lifelong learning discourse in Korea that facilitates evolution of the goals of older adult learners. Diplomatism has been regarded as a driving force of the expansion of public education through desire to earn a higher diploma [47]. In East-Asian countries, which are under the influence of the cultural values of Confucianism, diplomatism is common: people pay respect to educated people and expect their children to get higher diplomas. In Taiwan, for example, such a value often attracts older adults to pursue higher education degrees [20]. On the other hand, lifelong learning has been used as a concept of education reform in Korea since its introduction in 1995 and has been used as a direction of policies of the central and local governments since enactment of the Lifelong Education Act in 2008. The lifelong learning discourse formed during this process made people learning throughout life for granted [48]. We explain that it is the collective expectations that made learning “addictive” for Ahn1, that guided uneducated Soo to her status as a college graduate, and that led Chung to dream of entering graduate school at the age of 74.

## 5. Conclusions

In this paper, we examined the practice of active aging through a case study of older adult learning at evening schools in Korea. This study found themes that describe the experiences of research participants who are studying for equivalency examinations at evening school: overcoming limited education, taking the equivalency examination, and evolving goals. The first theme represents needs derived from older adults’ limited education and goals to meet them, the second describes how they encountered evening schools and took the equivalency examinations, and the third denotes development of goals after attending evening school and passing the equivalency examination. 

We discussed the findings under the topics of needs, system, and goals. Based on the discussion, we argue that lifelong learning is essential to active aging in that the needs of older adult learners are oriented instrumentally, the learners actively delay immediate gratification to satisfy those needs, participation in learning by older adults is an active agentic behavior in systemic change, and older adult learners are pulled toward goals they set actively. Furthermore, to compensate for the shortcomings of the conventional approach to older adult learning through psychological constructs, we added sociological analysis: life course approach to needs, understanding the system as interaction between structure and agency, and explaining the social embeddedness of goals in terms of collective expectation. We propose that a joint psychological and sociological approach could help explain the multifaceted nature of older adults’ participation in learning.

One of the limitations of this study was that small number of research participants that could not represent the experience of all older adults in evening school. Another limitation was the insufficiency of data collection about evening schools, which was crucial to understand the context of the experience of older adult learners. Future study can apply our methodological framework to obtain qualitative details about older adults’ learning experiences at other educational institutions. Despite these limitations, this study contributed to psychosocial interpretation of older adult participation in learning as a combination of psychological needs and socially embedded goals. In addition, this study has sociocultural impact in that it provides insights to understand older adults’ increasing needs for formal recognition of their education and participation in higher education.

## Figures and Tables

**Figure 1 ijerph-18-09232-f001:**
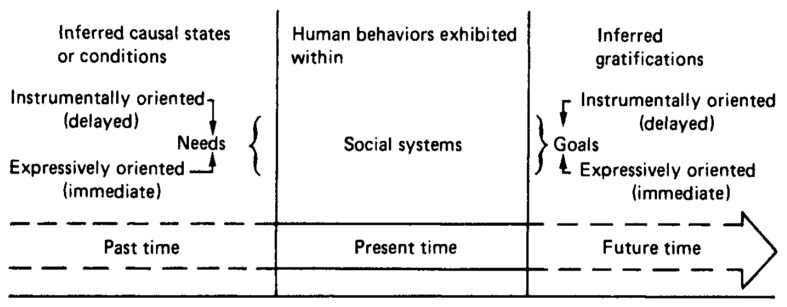
Londoner’s needs → social system → goal gratification model of older adults’ educational participation [22]. Reproduced with permission from Londoner, Carroll A., Sherron, Ronald H., Lumsden, D. Barry, *Introduction to Educational Gerontology*; published by Hemisphere Publishing Corporation: London, UK, 1990.

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
