# Peer review of "A Case Study of Active Aging through Lifelong Learning: Psychosocial Interpretation of Older Adult Participation in Evening Schools in Korea"

_ijerph, 2021, doi:10.3390/ijerph18179232_

Round 1

Reviewer 1 Report

Dear authorship

He is congratulated on the research study carried out and his interesting contribution to the scientific community.

We consider it of interest to advise you to refer the study to journals focused on the educational framework and its experiences. We are convinced that it will have a positive reception.

  1. Title and summary (clarity and structure.

The version of the title of the manuscript is concise, informative, and includes a number of possible identifying terms framing the theme that is developed in the document object review. Regarding the content, it is advisable to review the expression and link the presentation with the main objective of the study.

The authorship attends the indications on the editor's presentation and format.

  1. Relevance of the topic / Originality of the work / Literature review.

Again, we transfer our congratulations to the authorship for the theme developed and the innovative approach of the study. Originality and literature review, serious and well-founded. In this section, it is recommended to incorporate a more complete vision of the educational system, the national and international regulations regarding permanent education, and not only lifelong education. And incorporate basic notions about the study of cases from the anthropological perspective and the Chicago's School that links with the sociological, psychological, and educational conceptions that the authorship collects.

It is recommended to alternate the expression "realistic and experiential" of the literal quotes presented with a technical language appropriate in a scientific publication -participating researcher, qualitative paradigm, sociocritical model-.

  1. Methodological rigor / Research instruments.

The methodology described, for case study research works, is clear and concise, allowing its replication, by other experts. Methodological rigor is present in the design and instruments, moving to data analysis tools.

As for advice and proposals for improvement, before referring to other journals, the authorship is advised to incorporate into the manuscript a table with the "Variables, Categories, and Indicators" as well as a schedule or figure with the research design. It would contribute to the reader, systematicity, and clarity of the study.

  1. Results / Discussion / Conclusions.

The research results maintain the rigor and scientific exposition, denominator report common. It is adjusted to the achievement of the main objective of the investigation. It coincides with previous sections by suggesting a review of the structure and incorporating figures and tables that respond to the methodological structure -variables, categories, and indicators of the qualitative study- formatted according to the criteria of the journal to which the manuscript is sent.

It is advisable to incorporate a section of evaluation and self-evaluation of the process and sociocultural impact of the study and the monitoring of the sample.

Yours sincerely

Author Response

Reviewer 1

Comment

Response

1. Title and summary (clarity and structure.

The version of the title of the manuscript is concise, informative, and includes a number of possible identifying terms framing the theme that is developed in the document object review. Regarding the content, it is advisable to review the expression and link the presentation with the main objective of the study.

The we havehip attends the indications on the editor's presentation and format.

According to the advice, we have edited the Abstract to clearly link the content with the main objective.

2. Relevance of the topic / Originality of the work / Literature review.

Again, we transfer our congratulations to the we havehip for the theme developed and the innovative approach of the study. Originality and literature review, serious and well-founded. In this section, it is recommended to incorporate a more complete vision of the educational system, the national and international regulations regarding permanent education, and not only lifelong education. And incorporate basic notions about the study of cases from the anthropological perspective and the Chicago's School that links with the sociological, psychological, and educational conceptions that the we havehip collects.

It is recommended to alternate the expression "realistic and experiential" of the literal quotes presented with a technical language appropriate in a scientific publication -participating researcher, qualitative paradigm, sociocritical model-.

According to the recommendation of “a more complete vision of the educational system”, we have added accounts about the role of evening school and adults’ equivalency examination in the Korean education system and national permanent education to section 2.2.

According to the recommendation, we have incorporated basic notions about the study of cases from the anthropological perspective and the Chicago’s School in section 2.1.

According to the recommendation to alternate the expression of literal quotes “with a technical language appropriate in a scientific publication”, we have revised two literal quotes in Results section 3.1 and 3.3.

3. Methodological rigor / Research instruments.

The methodology described, for case study research works, is clear and concise, allowing its replication, by other experts. Methodological rigor is present in the design and instruments, moving to data analysis tools.

As for advice and proposals for improvement, before referring to other journals, the we havehip is advised to incorporate into the manuscript a table with the "Variables, Categories, and Indicators" as well as a schedule or figure with the research design. It would contribute to the reader, systematicity, and clarity of the study.

Regarding the advice for “systematicity and clarity of the study”, we have added an account of research design to section 2.1.

4. The research results maintain the rigor and scientific exposition, denominator report common. It is adjusted to the achievement of the main objective of the investigation. It coincides with previous sections by suggesting a review of the structure and incorporating figures and tables that respond to the methodological structure -variables, categories, and indicators of the qualitative study- formatted according to the criteria of the journal to which the manuscript is sent.

It is advisable to incorporate a section of evaluation and self-evaluation of the process and sociocultural impact of the study and the monitoring of the sample.

Regarding the advice to incorporate a section of evaluation, we have added a section regarding evaluation of the process, particularly on research limitations, and the sociocultural impact of the study to the Conclusion section.

Reviewer 2 Report

The topic of this manuscript, lifelong learning, is an important issue for improving knowledge in active aging.

  1. It is recommended to add a research design to the title to highlight the uniqueness of the research.
  2. It is recommended to explain that what kind of questions are asked as structured in semi-structured interviews.
  3. Please indicate how many researchers conduct interviews and data analysis, and how to deal with the internal consistency of the researchers.
  4. It is recommended to mark which results are from "documentation review".
  5. Since "theory-driven code development" is used for data analysis (page 3, Line 132), the role of theory cannot be seen in the results. In addition, it is also difficult to see how "instrumentally or expressively oriented needs and goals"(page 3, Line 137-138) is applied to data analysis and how to identify three thematic categories based on "instrumentally or expressively oriented needs and goals".
  6. "Three thematic categories" is more like the process/stage of the participants in the evening school. It is recommended to subtitle important findings in each thematic category to echo Boyatzis' theories. For example, an inferiority complex seems to be the need to drive the research participants to attend evening school. Another example maybe is the sources of information about evening school. And so on.
  7. According to the manuscript, "theory" is not so much the basis of data analysis, it is more like the framework of discussion. Please clarify.
  8. Please add a paragraph to specify the research limitations.
  9. It seems no description of research ethical concern and IRB certification number.

Author Response

R2

Comment

Response

1. It is recommended to add a research design to the title to highlight the uniqueness of the research.

To incorporate the research design into the title, we have revised the title to “A case study of active aging through lifelong learning: Psychological interpretation of older adult participation in evening schools in Korea

2. It is recommended to explain that what kind of questions are asked as structured in semi-structured interviews.

In accordance with the recommendation, we have added interview questions used in the study to section 2.4.

3. Please indicate how many researchers conduct interviews and data analysis, and how to deal with the internal consistency of the researchers.

According to the comment, we have added accounts how many researchers conducted data collection and analysis and how to deal with the internal consistency of the authors to sections 2.4 and 2.5.

4. It is recommended to mark which results are from "documentation review".

According to the recommendation, we have inserted a section into section 2.5 in order to specify results of documentation review.

5. Since "theory-driven code development" is used for data analysis (page 3, Line 132), the role of theory cannot be seen in the results. In addition, it is also difficult to see how "instrumentally or expressively oriented needs and goals"(page 3, Line 137-138) is applied to data analysis and how to identify three thematic categories based on "instrumentally or expressively oriented needs and goals".

According to the comment, we have added brief accounts about the role of theory in data analysis to section 2.5.

6. "Three thematic categories" is more like the process/stage of the participants in the evening school. It is recommended to subtitle important findings in each thematic category to echo Boyatzis' theories. For example, an inferiority complex seems to be the need to drive the research participants to attend evening school. Another example maybe is the sources of information about evening school. And so on.

In accordance with the recommendation, we have added subtitles of 3.1, 3.2, and 3.3 to highlight important findings.

7. According to the manuscript, "theory" is not so much the basis of data analysis, it is more like the framework of discussion. Please clarify.

Londoner’s temporal lifeline model composed of past time-present time-future time was used as the basis of data analysis. Thus, the results section themes were drawn from the theory-based data analysis. Furthermore, as you mentioned, the theory provided the framework of discussion.

8. Please add a paragraph to specify the research limitations.

According to the comment, we have added a paragraph of research limitations to the conclusion section.

9. It seems no description of research ethical concern and IRB certification number.

According to the comment, we have added the relevant accounts to the end of Conclusion section. IRB deliberation of the study was exempted by National Institute for Lifelong Education and we submitted relevant documents to the journal editorial board secretariat.

Reviewer 3 Report

Check references style, check number 33. Add more references, specially from the last 3-5 years.

The evening schools selected needs to be contextualized, is not the same rural than urban areas in Korea.  

5 informants was limited and shallow, it must be added as a limitation. Could the authors be found similar results with a big sample, with a group discussion, or by phone interviews? What future studies need to focus after this study?

The conclusion needs to anwer the title, is not clear: what is the Psychosocial interpretation of older adult participation? Considering the sample maybe is older-adults, not adults.

Author Response

R3

Comment

Response

1.Check references style, check number 33. Add more references, specially from the last 3-5 years.

According to the comment, we have corrected reference 33 and added recently-produced references to section Introduction, 4.1, and 4.2.

2. The evening schools selected needs to be contextualized, is not the same rural than urban areas in Korea.  

According to the comment, we have added a description of the evening schools selected to section 2.2.

3. 5 informants was limited and shallow, it must be added as a limitation. Could the we have be found similar results with a big sample, with a group discussion, or by phone interviews? What future studies need to focus after this study?

To address this limitation, we have added a paragraph of research limitations and future study suggestions to the conclusion section.

4. The conclusion needs to answer the title, is not clear: what is the Psychosocial interpretation of older adult participation? Considering the sample maybe is older-adults, not adults.

According to the comment, we have revised the conclusion to clarify the “psychosocial interpretation” statement.

Round 2

Reviewer 1 Report

Dear authors, The contributions in relation to the theoretical and methodological framework made in your study are considered acceptable. All the suggestions made have been included. Likewise, the bibliography provided completes the study and the citation standards have been rigorously reviewed. Therefore, it is considered that the study can be admitted to be published in the journal. The manuscript is at the discretion of the editors. Congratulations on your research. Best greeting 

This manuscript is a resubmission of an earlier submission. The following is a list of the peer review reports and author responses from that submission.